# Tunable self-healing of magnetically propelling colloidal carpets

Helena Massana-Cid[1,7], Fanlong Meng [2,3,7], Daiki Matsunaga [3,4], Ramin Golestanian [2,3] & Pietro Tierno [1,5,6]

The process of crystallization is difficult to observe for transported, out-of-equilibrium systems, as the continuous energy injection increases activity and competes with ordering. In emerging fields such as microfluidics and active matter, the formation of long-range order is often frustrated by the presence of hydrodynamics. Here we show that a population of colloidal rollers assembled by magnetic fields into large-scale propelling carpets can form perfect crystalline materials upon suitable balance between magnetism and hydrodynamics. We demonstrate a field-tunable annealing protocol based on a controlled colloidal flow above the carpet that enables complete crystallization after a few seconds of propulsion. The structural transition from a disordered to a crystalline carpet phase is captured via spatial and temporal correlation functions. Our findings unveil a novel pathway to magnetically anneal clusters of propelling particles, bridging driven systems with crystallization and freezing in material science.

[1] Departament de Física de la Matèria Condensada, Universitat de Barcelona, 08028 Barcelona, Spain. [2] Max Planck Institute for Dynamics and Self-Organization (MPIDS), 37077 Göttingen, Germany. [3] Rudolf Peierls Centre for Theoretical Physics, University of Oxford, Oxford OX1 3PU, UK. [4] Division of Bioengineering, Graduate School of Engineering Science, Osaka University, 5608531 Osaka, Japan. [5] Institut de Nanociència i Nanotecnologia, Universitat de Barcelona, Barcelona 08028, Spain. [6] Universitat de Barcelona Institute of Complex Systems (UBICS), Universitat de Barcelona, Barcelona 08028, Spain. [7] These authors contributed equally: Helena Massana-Cid, Fanlong Meng. Correspondence and requests for materials should be addressed to R. G. (email: ramin.golestanian@ds.mpg.de) or to P.T. (email: ptierno@ub.edu)

In colloidal science, crystalline order is usually obtained from equilibrium self-assembly, when a system spontaneously forms an organized phase due to specific inter-particle interactions. This general phenomenon has proven to be simple, robust, and scalable, all appealing features that make colloidal crystals ideal candidates for photonic band gap materials, optical switches, or sensors. Investigating the assembly process of tunable colloidal systems may also shed light on fundamental mechanisms underlying melting[1] and crystallization[2], which are general phenomena occurring in a broad range of systems at different length scales[3].

Recent trends in the field are now shifting the focus toward structure formation in systems driven out-of-equilibrium by external fields or forces. Examples are widespread and include the assembly induced by external electric[4,5], magnetic[6,7], optic[8] fields, or the organization of active particles[9–13]. However, an important challenge that still remains to be tackled is whether it is possible to realize perfect crystalline lattices starting from a disordered collection of driven particles, where the individual units display a net propulsive dynamics. Such a feature would be important not only for practical means, i.e., to rapidly realize periodic system on the visible wave-length, but it will also provide a starting point for understanding the fundamental mechanisms behind crystal formation in driven or active out-of-equilibrium systems. We also note that a crystallization process induced by a few active dopants in a bath of passive particles has been demonstrated recently in computer simulation studies[14,15]. Recent experiments with active particles report melting[16], clustering[10,13], or interstitial dynamics[17] but not annealing.

Here we advance in this field by demonstrating a novel field-induced annealing process where an ensemble of propelling particles is able to rapidly form a crystalline lattice upon magnetic command. In particular, we study the dynamics of magnetic colloidal rotors that are assembled into flat, two-dimensional propelling carpets due to external magnetic field modulations. In a previous work[18], we demonstrate that the carpet could be used as an efficient drug-delivery vector for transporting biological cells across its surface. Beyond its technological applications, here we report the discovery that such carpet could be used as a general model system for nonequilibrium crystallization process, by demonstrating a self-healing process that can be controlled by an external magnetic field. We show that by increasing the field amplitude the carpet displays a tread-milling dynamics, where particles detach from the back of the carpet, travel across the lattice, and reattach at the moving front. The dynamic phases of the system may be understood from the delicate balance between magnetism and hydrodynamics. This regrowth process represents a novel annealing process that enables to rapidly regenerate colloidal structures upon magnetic command.

## Results

### Assembly and propulsion of colloidal carpets

We assemble the colloidal carpets from a dispersion of paramagnetic colloids having radius $a = 1.4\,\mu m$ and subjected to time-dependent external fields. The particles are initially dispersed in water and two-dimensionally confined above a glass substrate due to the balance between gravity and electrostatic interactions (see Methods). Figure 1a–f illustrate the complete process. To realize a propelling carpet we start to apply a rotating magnetic field circularly polarized in the plane of the substrate $(\hat{x}, \hat{y})$, $\boldsymbol{B}_1(t) = B_0[\cos(2\pi ft)\hat{x} - \sin(2\pi ft)\hat{y}]$, where $B_0$ is the field amplitude and $f$ is its frequency. We use a narrow frequency range $f \in [20, 100]\,Hz$, far away from resonance frequency (~400 KHz) as reported in the past[19]. For sufficiently high frequencies, the rotating field induces attractive dipolar interactions that are isotropic when time-averaged[18]. The dipolar interaction between two equal dipoles $\boldsymbol{m}_{i,j}$ at a distance $\boldsymbol{r}_{ij} = \boldsymbol{r}_i - \boldsymbol{r}_j$ is given by

$U_m = -\mu_0/4\pi\{[3(\boldsymbol{m}_i \cdot \boldsymbol{r}_{ij})(\boldsymbol{m}_j \cdot \boldsymbol{r}_{ij})/r^5] - (\boldsymbol{m}_i \cdot \boldsymbol{m}_j)/r^3\}$, and becomes maximally attractive (repulsive) for particles with magnetic moments parallel (normal) to $\boldsymbol{r}_{ij}$. Performing a time average of the potential gives an effective attractive interaction in this plane $\langle U_m \rangle = -\mu_0 m^2/[8\pi(x+y)^3]$. Thus, a random dispersion of particles that would otherwise perform simple Brownian motion, is forced to assemble into a compact cluster, as shown in Fig. 1a, b. Under the rotating field, the cluster is also observed to perform a spinning motion around its center, since the rotating field applies a magnetic torque, $\boldsymbol{T}_m \sim B_0^2$, due to the finite internal relaxation time of the particle, see Methods. Once the cluster is formed, propulsion of the carpet is obtained with the following strategy. The rotating in-plane field is transformed in a more complex modulation composed by a field that rotates in the $(\hat{x}, \hat{z})$ plane plus an oscillating component along the $\hat{y}$ direction with a different frequency $f_y$. The full expression is given as $\boldsymbol{B}_2(t) = B_0[\cos(2\pi ft)\hat{x} + \sin(2\pi f_y t)\hat{y} - (B_z/B_0)\sin(2\pi ft)]$, where $f_y = f/2$ (see Fig. 1b, c). The rotating field in the perpendicular plane, $(\hat{x}, \hat{z})$, is used to induce a magnetic torque on the individual particles, which now rotate close to the glass substrate. As the particles are hydrodynamically coupled to the plane[20], this rotational motion is converted into a net translational one. As a consequence, all the cluster elements are now a collection of microscopic rotors that make the whole carpet translating at a constant average speed as shown in Fig. 1e, f (see also Supplementary Movie 1 in the Supplementary Information). On the other hand, the carpet is kept stable by the oscillating component $B_y$ that avoids lateral separation of the rotors due to attractive dipolar interactions.

The carpet speed $v_c$ can be varied mainly by changing the field amplitude $B_0$ or the number of rotors $N$. In the latter case it becomes constant above $N \sim 300$, which corresponds to a carpet area $S \sim 1800\,\mu m^2$. Although a previous work[18] analyzed the dependence of $v_c$ on $N$, here we focus on much larger carpets ($N \geq 1000$ particles) and use only $B_0$ to tune $v_c$. Moreover, previously the amplitude of the perpendicular field $B_z$ was always kept below a threshold value, in order to stabilize the structure confined in two dimensions[18]. In contrast, here we show a series of remarkable new phenomena that emerge when the propelling carpet is forced to extend toward the third dimension by raising $B_z$. We also find that, for a certain range of field amplitudes close to the transition region in Fig. 1g, our method allows the relative motion of two lattice planes of colloidal particles. This phenomenon could offer new possibilities in the study of frictional effects at the microscale in the presence of hydrodynamic lubrication.

In Fig. 1g we show the different dynamic states observed in the $(B_0, B_z)$ space, where stable carpets with defects (vacancies) are found at low perpendicular field $B_z$ ("2D carpet"). Increasing $B_z$ forces the carpet to fold into a membrane in the third dimension. However, competition with gravity breaks the monolayer into a series of separate, rolling chains ("broken carpet"), see also Supplementary Movie 2 and Supplementary Movie 3 in the Supplementary Information. Between both dynamic states, we found a stationary phase where particles are transported above the carpet, while the entire structure continues to propel close to the plane ("tread-milling"). We explain the different dynamic phases observed in Fig. 1g by considering the balance between magnetic and hydrodynamic interactions. For a pair of paramagnetic particles located at a distance $\boldsymbol{r}$, the dipole–dipole interaction averaged over one period of the rotating field is given by

$$\langle U_m \rangle = \frac{(V\chi B_0)^2}{4\pi\mu_0 r^3}\left[1 - \frac{B_z^2}{B_0^2} - \frac{3}{2}\left(1 - \frac{B_z^2}{B_0^2}\right)\sin^2\theta\right], \qquad (1)$$

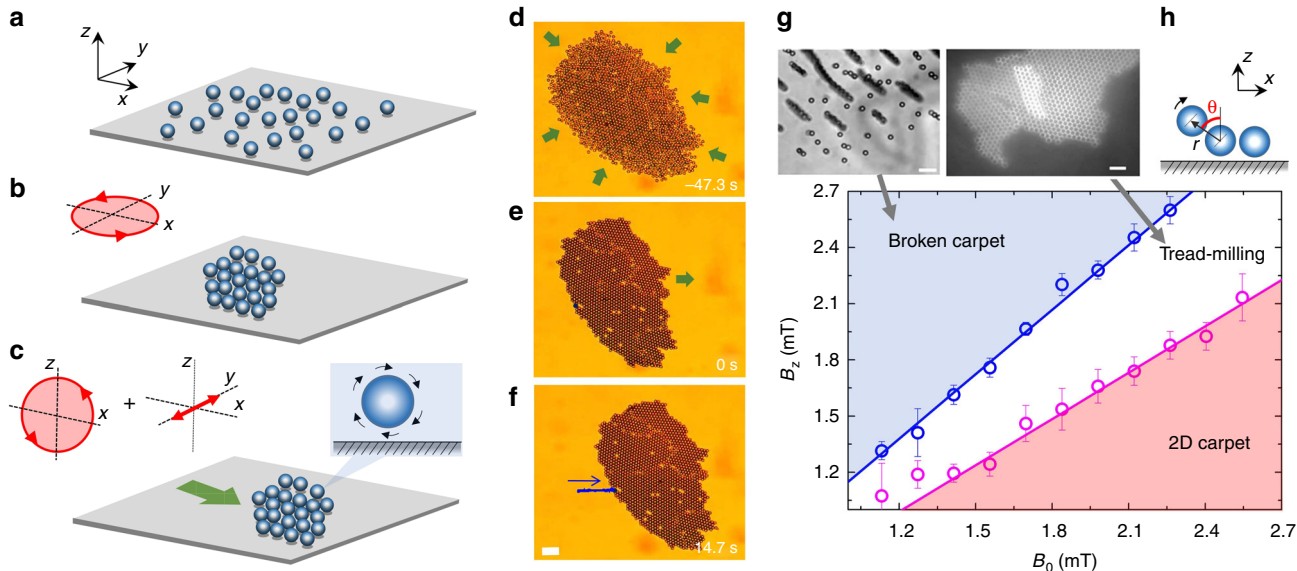

**Fig. 1** Realization of a propelling carpet and out-of-equilibrium phase diagram. **a**–**c** Sequence of schematics showing the formation of a carpet due to a rotating field in the plane $(\hat{x}, \hat{y})$ (**a**, **b**) and its propulsion induced by a field rotating in the perpendicular plane $(\hat{x}, \hat{z})$ plus an oscillating component along the $\hat{y}$ direction (**b**, **c**). **d**–**f** Corresponding optical microscope images of a carpet assembled by a rotating field with amplitude $B_0 = 1.6$ mT and frequency $f = 40$ Hz (**d**, **e**), and transported by a field $\boldsymbol{B}_2$ with amplitudes $B_0$, $B_z = 1.7$ mT, and frequencies $f_x = f_z = f$, $f_y = f/2$ (**e**, **f**). In image **f**, the trajectory of one particle is superimposed. Scale bar is 20 μm, see Supplementary Movie 1 in the Supplementary Information. **g** Diagram showing the different assembled structure observed in the $(B_0, B_z)$ plane. The pink region denotes stable propelling structures ("2D carpet"), whereas the white region refers to the situation where the paramagnetic colloids flow above the monolayer ("tread-milling"). The blue region indicates the formation of three-dimensional structures ("broken carpet"). Scattered points are experimental data and continuous lines result from linear stability analysis as described in the text. Error bars are obtained from the statistical average of different experiments. **h** Schematic illustrating the definition of the angle $\theta$ and position vector **r**. Videos illustrating the tread-milling motion and formation of broken carpets (Supplementary Movie 2 and Supplementary Movie 3) are in the Supplementary Information. Scale bars of the experimental images at the top are 20 μm

where $V = (4/3)\pi a^3$, $\chi = 0.4$ is the magnetic volume susceptibility and $\mu_0 = 4\pi \times 10^{-7}$ Hm. We then assume that the hydrodynamic interactions between the two particles can be cast in terms of an effective energy potential given by

$$U_{\mathrm{h}} = 3\pi \eta a^3 \omega_c \theta, \qquad (2)$$

where $\eta$ the solvent viscosity (water), $\theta$ is the angle between the two particles (see Fig. 1h). Here $\boldsymbol{\omega}_c = B_0 B_z \chi \tau_r \omega [6\eta\mu_0(1 + \tau_r^2 \omega^2)]\hat{y}$ is the particle angular velocity, which is in general lower than the driving one, $\omega = 2\pi f$, and $\tau_r$ is the magnetic relaxation time of the paramagnetic colloids (see Methods). By combining the contributions from the magnetic dipole–dipole interaction and hydrodynamics, the effective total energy of the colloid that is located at the edge becomes $U_{\mathrm{tot}} = U_{\mathrm{m}} + U_{\mathrm{h}}$. This leads to the dynamic equation for the angle $\theta$ as

$$\dot{\theta} = -\frac{1}{4\zeta a^2} \frac{\partial U_{tot}}{\partial \theta}, \qquad (3)$$

where $\zeta$ is the friction coefficient, see Methods for a detailed derivation.

From Eq. (3), the condition for obtaining a stable $\theta$ is given by:

$$|\sin 2\theta| = \left| \frac{6 B_0 B_z}{\chi (B_0^2 - B_z^2)} \frac{\tau_r \omega}{(1 + \tau_r^2 \omega^2)} \right| \leq 1. \qquad (4)$$

Equation (4) allows us to understand how the particle behavior changes at the rear edge of the carpet by varying the ratio of field amplitudes, $B_z^2/B_0^2$. (1) For small $B_z^2/B_0^2$, there is a stable solution $\theta \sim \pi/2$ corresponding to a structure where the colloid at the rear edge tends to lie in the $(\hat{x}, \hat{y})$ plane; (2) for moderate values of $B_z^2/B_0^2$, there is no stable solution of $\theta$ and then the

colloid at the rear edge of the carpet is transported toward the front edge ("tread-milling"); (3) for large $B_z^2/B_0^2$, there is a solution $\theta \sim 0$ and the colloid at the rear edge "stands up," giving rise to a structure in the $(\hat{y}, \hat{z})$ plane ("Broken carpet"). As shown in Fig. 1g, we find that the developed model is in excellent agreement with the experimental data with the following conditions: $B_z/B_0 < 0.84$ for carpet confined in the $(\hat{x}, \hat{y})$ plane, and $B_z/B_0 > 1.19$ for a disk in $(\hat{y}, \hat{z})$ plane. The fit to the data are obtained by using as sole adjustable parameter the relaxation time $\tau_r \simeq 10^{-4}$ s, which is in agreement with the value found in separate experiments[18].

## Discussion

We next explore in more details the tread-milling phase and how it induces a regeneration of the propelling carpet. By increasing the amplitude of $B_z$, the particles are forced to detach from the back of the carpet and are transported above the surface by the hydrodynamic flow field. The detached particles move faster than the underlying monolayer, but do not leave the structure when they reach the end, as they remain strongly attracted there by dipolar interactions. Due to this attraction, the particles follow one of the crystallographic axes of the underlying lattice during transport. As shown in Fig. 2a, b, three different situations can be observed, illustrated also in Supplementary Movie 4 in the Supplementary Information. The particles can be adsorbed in the monolayer by filling vacancies that encounter during their excursion, they can change direction when they reach a grain boundary that merges domains with different crystalline orientations, or they can reach the moving front and reattach there. In the latter case, the particles form a growing interface that replicates the carpet wedge and shows no defect or vacancies. This

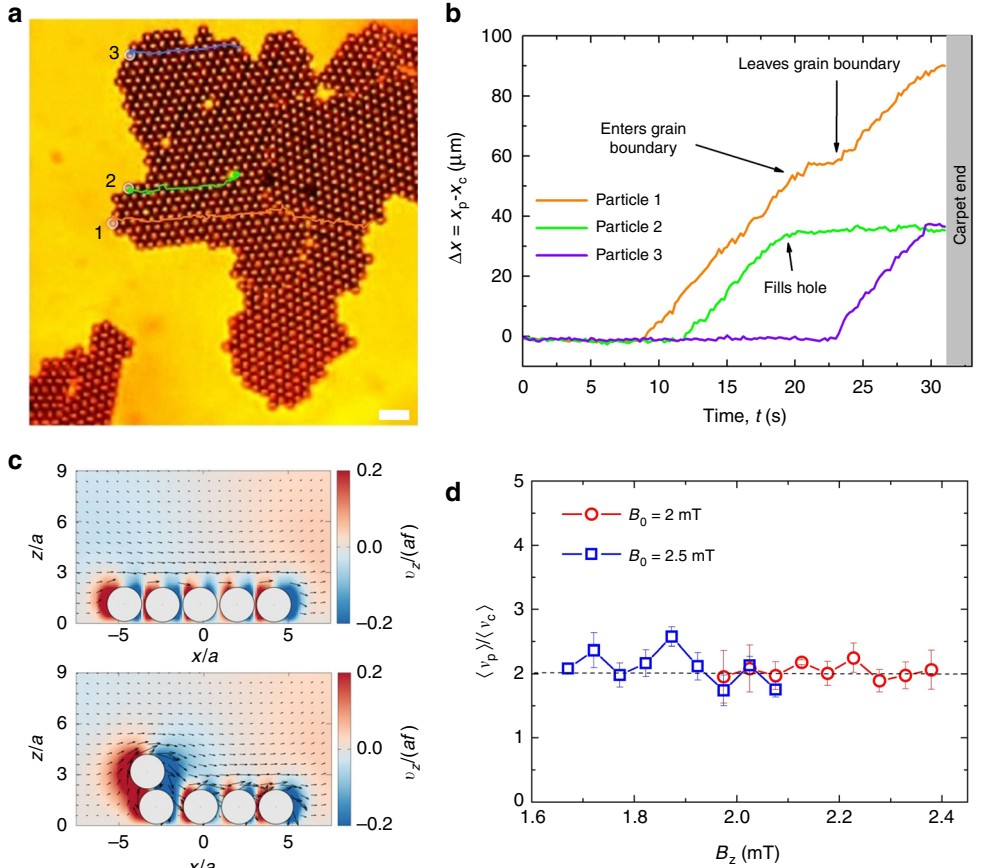

**Fig. 2** Tread-milling dynamics: particle flow and speed. **a** Optical microscope image of a traveling carpet with the superimposed trajectories of three particles illustrating the different situations encountered. The carpet translates under the action of a dynamic field with amplitudes $B_0 = 2$ mT, $B_z = 2$ mT, and frequency $f = 40$ Hz. The scale bar is 20 μm; see Supplementary Movie 4 in the Supplementary Information. **b** Relative distance, $x_p - x_c$ of the three paramagnetic colloids vs. time. Particle 1 enters and leaves a grain boundary above the carpet changing direction but not speed. Particle 2 fills an hole while particle 3 reaches the carpet edge. **c** Calculated flow field around the rotating colloids using boundary element simulation (see Methods). The vectors show the flow direction, while the color map illustrates the $\hat{z}$-component of velocity. The image shows the process of detachment of one particle at the back of the carpet. **d** Average particle speed $\langle v_p \rangle$ above the colloidal monolayer normalized to the average carpet velocity $\langle v_c \rangle$. Error bars are obtained from the statistical average of different experiments

process can be also appreciated from Supplementary Movie 2 in the Supplementary Information. Furthermore, we calculate the hydrodynamic flow field generated by the particles using a boundary element simulation technique; see Fig. 2c for the corresponding velocity profile and the Methods section for more details. The moving carpet generates a cooperative chiral flow field, which advects continuously the monolayer of particles in a similar way as a hydrodynamic conveyor belt. Although the flow velocity is small below the carpet given the short distance with the surface, it increases significantly close to the edge of the structure and thus forces particle detachment there. The average speed of the particles traveling above the carpet $\langle v_p \rangle$ is twice the carpet speed $\langle v_c \rangle$ as shown in Fig. 2d. This result, which is consistent with the kinematics of rolling motion, is confirmed in the numerical simulation.

The tread-milling phase generates a net colloidal flow above the carpet allowing to completely rebuild its structure in a relatively short time scale. This occurs as the growing front of the carpet crystallizes in a perfect periodic lattice of colloidal rotors, while it smoothen the edge of the structure. The process is illustrated in Fig. 3a, where an initially disordered structure composed by 1045 particles is propelled at a speed of $\langle v_c \rangle = 5.7$ μm s$^{-1}$ and after $t = 2$ min it has regrown with a perfect crystalline

order. We analyze this process in terms of bond-orientational correlation function

$$g_6(r_{ij}) = \left\langle \Psi_6^*(r_i)\Psi_6(r_j) \right\rangle, \tag{5}$$

where $\Psi_6(r_j) = \left| \frac{1}{N_b} \sum_{k=1}^{nn} e^{6i\theta_{jk}} \right|$, $N_b$ is the number of neighboring particles $k$, and $\theta_{kj}$ is the angle between a fixed axis and the bond joining particles $k$ and $j$. The initial disordered state results from the assembly of the carpet due to the rotating field and gives rise to a hexatic structure, with an algebraic decay of $g_6(r) \sim r^{-\eta_6}$ that characterizes quasi-long-range order. Here we find an exponent $\eta_6 \sim 0.3$, which is higher than the threshold value 1/4 as predicted by the KTNHY theory[1] of melting in two- dimensional (2D) at equilibrium. As time advances, the carpet front recrystallizes and the nominal exponent of the correlation function decreases (to values as small as $\eta_6 \sim 0.1$). In this nonequilibrium situation, all rotors are characterized by six nearest neighbors and long-range order develops.

We then monitor the ability of the monolayer to recrystallize or heal itself by varying the carpet speed $v_c$ via the external field amplitudes $B_0$ and $B_z$. As shown in Fig. 3c, in order to keep the carpet within the tread-milling phase, we need to simultaneously vary both amplitudes, which allows us to increase $v_c$ from 2.5 to ~

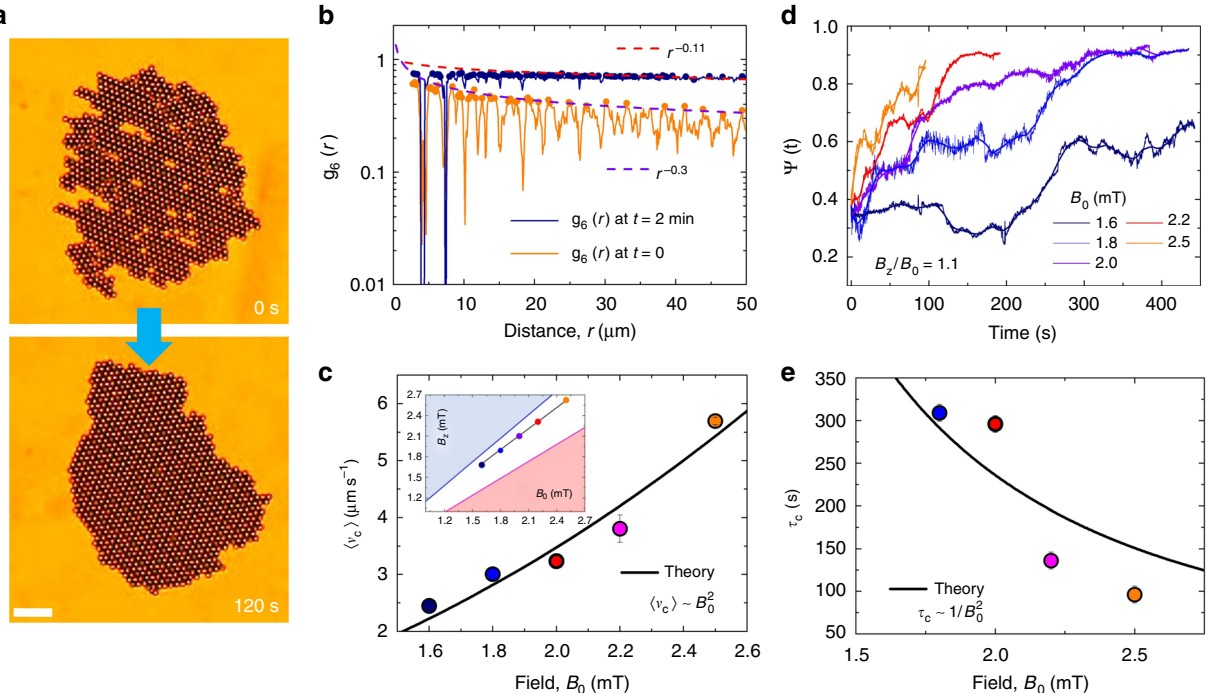

**Fig. 3** Annealing of magnetic carpet. **a** Experimental images showing an initial polycrystalline carpet ($t = 0$s, top) realized with an in-plane rotating field ($B_0 = 2.5$ mT, $f = 40$ Hz), and a monocrystalline carpet obtained after $t = 2$ min of propulsion ($B_0 = 2.5$ mT, $B_z = 2.7$ mT, $f = 40$ Hz). The scale bar is 20 μm, see Supplementary Movie 5 in the Supplementary Information. **b** Bond-orientational correlation functions $g_6(r)$ of the two carpets with superimposed corresponding peaks as solid disks and fits to an algebraic dependence $g_6(r) \sim r^{-\eta_6}$ as dashed lines. The exponent $\eta_6$ can be used to distinguish the crystalline phase ($\eta_6 > 1/4$) from the hexatic one[3]. **c** Average carpet speed $\langle v_c \rangle$ vs. field amplitude $B_0$ for different values of $B_z$. The continuous black line is a fit to the data using Eq. (16), with fixed parameters $l = 3.2$ μm, $\chi = 0.4$, $\omega = 251.3$ rads$^{-1}$ and $h = 2.6$ μm. The small inset indicates the corresponding location of the experimental points in the diagram in Fig. 1g ($B_0 \sim B_z$ is taken). Error bars are obtained from the statistical average of different experiments. **d** Bond-orientational order parameter $\Psi(t)$ vs. time $t$ for different values of $B_0$ (thus carpet speed $v_c$). In the continuous lines in the graph is obtained by averaging the experimental data over a discrete time window. All experiments are conducted for carpets having a similar size of ~1000 particles. **e** Annealing time $\tau_c$ vs. field amplitude $B_0$, with continuous black line is a fit to the data using Eq. (6)

6 μm s$^{-1}$. The carpet speed can be calculated by considering a triangular lattice of surface rotors, with lattice constant $l$ (see Methods section for more details). We can fit the experimental data of Fig. 3c with Eq. (16) using $l \sim 3.2$ μm with the magnetic relaxation time $\tau_r$ as an adjustable parameter. We obtain a value of $\tau_r = 0.3 \times 10^{-4}$ s that is consistent with the diagram in Fig. 1g.

The dynamics of the ordering process is then characterized via both the nominal time-dependent exponent $\eta_6$ of $g_6(r)$ (see Supplementary Fig. 1 in the Supplementary Information) and the global orientational order parameter averaged over all $N$ particles, $\Psi(t) = \frac{1}{N} \sum_{i=1}^{N} \Psi_{6,i}$ shown in Fig. 3d. In all cases we find that the increase in the carpet speed boosts the annealing process and allows the carpet to reach the crystalline phase more quickly. We confirm these observations by extracting the annealing time $\tau_c$ of the carpet as a function of the applied magnetic field (Fig. 3e). The time to eliminate a single defect in a carpet of area $L_x \times L_y$ can be approximated as $\tau_p \simeq L_y / v_p$, with the tread-milling speed given by $v_p \sim B_0^2 \chi \tau_r \omega a / [6 \eta \mu_0 (1 + \tau_r^2 \omega^2)]$. For a carpet with $N_d$ defects before annealing, the total annealing time can be approximated as the corresponding annealing time of one colloidal row along the $y$ axis, given by

$$\tau_c = n\tau_p \sim \frac{12 N_d L_y \eta \mu_0 (1 + \tau_r^2 \omega^2)}{L_x B_0^2 \chi \tau_r \omega}, \quad (6)$$

where $n = 2 N_d \, a / L_x$ is the number of the defects along one colloid row. For $L_x \sim L_y$, the annealing time can be simplified as

$\tau_c \sim \frac{12 N_d \eta \mu_0 (1 + \tau_r^2 \omega^2)}{B_0^2 \chi \tau_r \omega}$ and can be used to fit the experimental data shown in Fig. 3e.

We also investigate the scalability of the carpet formation (Supplementary Fig. 2 in Supplementary Information), propulsion (Supplementary Fig. 3 in Supplementary Information), and the tread-milling motion (Supplementary Fig. 4 in Supplementary Information). By varying the number of particles in the carpet, we find that the initial self-assembly process could lead to an ordered crystalline structure below $N = 150$ particles, as shown in Supplementary Fig. 2 in the Supplementary Information. Larger carpets assembled by the rotating field display grain boundaries and vacancies that require the tread-milling motion to crystallize. The carpet speed increases with the number of particles, reaching a saturation value above $N \sim 300$ particles, as shown in Supplementary Fig. 3 in the Supplementary Information. Also, we found that the tread-milling behavior can be observed for all the carpet size, and until filling completely the observation area of our experimental system for $N = 4000$ particles (see Supplementary Fig. 4 in Supplementary Information).

In active colloidal systems, increasing particle activity via the propulsion speed leads to an increase in the effective diffusion coefficient or effective temperature of the system, which we might expect to give rise to a reduction of the global order. Here we report exactly the opposite effect: the faster the propelling carpets are, the more rapidly they form ordered structures. Our novel results demonstrate the versatility of our

active magnetic system, as the governing interactions are based on a subtle interplay between hydrodynamics and magnetism

In conclusion, we investigate the collective dynamics of propelling magnetic carpets in a range of field parameters, where the colloidal structures are able to continuously transform into perfect crystalline lattices. The annealing process occurs due to the detachment of individual rotors from the back of the carpet, traveling above the carpet surface following a crystallographic direction, and either filling vacancies or reaching and rejoining the leading edge. We theoretically account for the out-of-equilibrium phase diagram by using a delicate balance between magnetism and hydrodynamics. The mechanism of the motility of our carpets is cooperative and is based on the rectification of the hydrodynamic flow generated by each rotor close to the bounding wall. Moreover, the use of external field allows us to steer and control the colloidal carpet and the corresponding flow of magnetic colloids that dictate the annealing process. From an application point of view, our annealing process could be easily extended to other types of recently engineered microscale particles with heterogeneous and functional properties[21–23], or used to entrap, transport, and release non-magnetic particles across the carpet surface[18]. All in all, our out-of-equilibrium colloidal model system allows us to investigate crystallization in transported systems and could thus provide deep insight for similar processes occurring in systems at different length and time scales.

## Methods

**Experimental details**. We use paramagnetic colloidal particles (Dynabeads M-270, Invitrogen) with diameter $d = 2.8\,\mu m$, density $\rho = 1.3\,g\,cm^{-3}$, and magnetic volume susceptibility $\chi = 0.4$. The particles are coated with surface carboxylic acid group with an active chemical functionality of $150\,\mu mol \times g^{-1}$ per particle. When dispersed in highly deionized water ($18.2\,M\Omega \times cm^{-1}$, MilliQ system), hydrogen ions ($H^+$) dissociate from such groups, leaving a negative charged surface and inducing the formation of a double layer. The solution containing the particles is introduced by capillarity in a rectangular microtube made of borosilicate glass (inner dimensions $0.1 \times 2.0\,mm$; CMC Scientific) that is immediately sealed. The particles sediment close to a glass plate, where they remain quasi-2D confined due to gravity, displaying a small thermal motion. The sample is placed in the center of a triaxial coil system arranged on the stage of a light microscope (Eclipse Ni, Nikon). External time-dependent magnetic fields are generated by passing an alternate current through the coils via a waveform generator (TGA1244, TTi) connected to different power amplifiers (AMP-1800, AKIYAMA, and BOP 20-10M, Kepco). The particle position and dynamics are extracted using digital videomicroscopy with a charge-coupled device camera (Scout scA640-74fc, Basler) working at 50 frames per second.

**Theoretical model**. Under a static external magnetic field $\mathbf{B}$, a paramagnetic colloid of radius $a$ acquires an induced moment $\mathbf{m} = V\chi\mathbf{B}/\mu_0$, where $V = 4\pi a^3/3$ is the particle volume. For a dynamic field, such as the field $\mathbf{B}_1(t)$ rotating in the $(\hat{\mathbf{x}}, \hat{\mathbf{z}})$ plane, the paramagnetic colloid experiences a finite magnetic torque[24]

$$\mathbf{T}_c = \frac{B_0 B_z V\chi\tau_r\omega}{\mu_0(1 + \tau_r^2\omega^2)}\hat{\mathbf{y}} \tag{7}$$

when the angular frequency of the magnetic field, $\omega = 2\pi f$, is comparable with the inverse of the magnetization relaxation time, $\omega\tau_r \sim 1$. Here, $\tau_r \sim 10^{-4}\,s$ for our paramagnetic colloids[18]. Thus, a colloidal particle suspended in a fluid of viscosity $\eta$ will rotate with angular velocity $\boldsymbol{\omega}_c < \omega$. The hydrodynamic interaction with the substrate induces a translational motion of the particle with velocity $v \sim \omega_c a$. In the case of translating carpet, the induced moment for the individual colloid is given by,

$$\mathbf{m} = \frac{4}{3\mu_o}\pi a^3\chi B_0\left(\cos\omega t, \sin\omega_2 t, -\frac{B_z}{B_0}\sin\omega t\right). \tag{8}$$

The energy due to the magnetic dipole–dipole interaction among the magnetized colloids is:

$$U_m = -\sum_{i,j\neq i}\frac{\mu_0\left[3(\mathbf{m}_i \cdot \mathbf{r}_{ij})(\mathbf{m}_j \cdot \mathbf{r}_{ij}) - \mathbf{m}_i \cdot \mathbf{m}_j r_{ij}^2\right]}{4\pi r_{ij}^5}, \tag{9}$$

where $\mathbf{r}_{ij}$ denotes the vector pointing from the $i^{th}$ colloid to the $j^{th}$ one. Due to the employed field strength, we will consider these interactions only at the level of

nearest neighbors. For two close particles as depicted in Fig. 1h, the magnetic dipole–dipole interaction averaged over one period of the rotating magnetic field can be explicitly expressed by Eq. (1), with the vector pointing from one colloid to the other as,

$$\mathbf{r} = r(\sin\theta\cos\phi, \sin\theta\sin\phi, \cos\theta),\ r \approx 2a. \tag{10}$$

Note that when $B_z \ll B_0$, the averaged magnetic dipole–dipole interaction is attractive in the $(\hat{\mathbf{x}}, \hat{\mathbf{y}})$ plane and the particles will form a carpet.

The flow field that the colloids at the edges experience can be approximately by a two-colloid model. Let us denote the colloid at the rear edge as colloid 1 and its nearest neighbor in $\hat{\mathbf{x}}$ direction as colloid 2. The rotating colloid 2 induces a flow field around it, which can be approximated by a rotlet. In this case, the flow velocity that colloid 1 (edge colloid) experiences is approximately:

$$\mathbf{v} = \frac{\mathbf{T} \times \mathbf{r}}{8\pi\eta r^3} = \frac{a\boldsymbol{\omega}_c \times \hat{\mathbf{r}}}{4} \tag{11}$$

where the distance between the colloids is taken as $r \approx 2a$. As a result, colloid 2 will rotate around 1. Or effectively, it can be considered that there is an effective torque acting on colloid 2 by the rotating colloid 1,

$$\mathbf{T}_h = 3\pi\eta a^3\boldsymbol{\omega}_c = \frac{\pi a^3 B_0 B_z\chi\tau_r\omega}{2\mu_0(1 + \tau_r^2\omega^2)}\hat{\mathbf{y}}. \tag{12}$$

We thus write an effective hydrodynamic potential as, $U_h = T_h\theta$ that gives rise to the total potential $U_{tot} = U_m + U_h$. The dynamic equation in terms of the orientation of the colloid, i.e., Eq. (3) can be expressed as:

$$\dot{\theta} = \frac{1}{4\zeta a^2}\frac{3(V\chi B_0)^2}{64\mu_0\pi a^3}\left(1 - \frac{B_z^2}{B_0^2}\right)\sin 2\theta - \frac{1}{4\zeta a^2}\frac{\pi a^3 B_0 B_z\chi\tau_r\omega}{2\mu_0(1 + \tau_r^2\omega^2)}. \tag{13}$$

Thus, the conditions for obtaining stable $\theta$ become:

$$\frac{B_z}{B_0} \leq \frac{-c + \sqrt{c^2 + 4}}{2}, \tag{14}$$

$$\frac{B_z}{B_0} \geq \frac{c + \sqrt{c^2 + 4}}{2}, \quad c = 6\tau_r\omega/[\chi(1 + \tau_r^2\omega^2)] \tag{15}$$

or alternatively Eq. (4) in the manuscript.

**Carpet speed**. Here we calculate the mean speed of a carpet composed by a triangular lattice of rotating paramagnetic colloids. An individual particle close to a surface and subjected to a magnetic torque $T_c\mathbf{e}_x$ acquires a propulsion speed $\mathbf{v}_0 = \frac{T_c a^2}{32\pi\eta h^4}\mathbf{e}_y$, where $a$ is the radius of the particle and $h$ its elevation from the surface. We consider the hydrodynamic interactions of the particle in presence of the surface and extend the calculations in ref. [25] to a 2D triangular lattice. If we consider that the rotating particles form a triangular lattice with lattice constant $l$, the particle position can be denoted by: $l(i\mathbf{u} + j\mathbf{v})$, $-N \leq i, j \leq N$, where $\mathbf{u} = (1, 0)$ and $\mathbf{v} = (1/2, \sqrt{3}/2)$ are the base unit vectors of the lattice. By treating each rotating colloid as a rotlet, the velocity of the colloid at the lattice center $(0, 0)$ is given by:

$$\mathbf{v} = \mathbf{v}_0 + \sum_{i=-N}^{N}\sum_{j=-N}^{N}\left(\frac{a}{h}\right)^{-2}\frac{3\varepsilon^2(i+j/2)^2}{\{1 + [(i+j/2)^2 + (\sqrt{3}/2)^2]\varepsilon^2\}^{5/2}}\mathbf{v}_0$$
$$\simeq T_c\left(\frac{a^2}{32\pi\eta h^4} + \frac{1}{4\eta l^2}\right)\mathbf{e}_y. \tag{16}$$

where $\varepsilon = \delta/2h$ with $\delta$ being the the center-to-center separation between consecutive colloids in the array.

**Boundary element simulation**. We describe the flow field $\mathbf{v}$ at a given point $\mathbf{x}$ using a boundary integral formulation [26], which is a surface integral on the particle surface as

$$v_i(\mathbf{x}) = -\frac{1}{8\pi\eta}\sum_m^M\int_{A_m}G_{ij}(\mathbf{x}, \mathbf{y})q_j(\mathbf{y})dA \tag{17}$$

where $\mathbf{G}$ is the Blake tensor[27], $A_m$ is the surface of $m$-th particle, and $\mathbf{q}$ is the viscous traction acting at a point $\mathbf{y}$ on the surface. Integrating the traction force $\mathbf{q}$ over a sphere surface gives the hydrodynamic force $\mathbf{F}_h$ and torque $\mathbf{T}_h$ acting on the particle. As each particle is in force- and torque-free conditions, these satisfy

$$\mathbf{F}_h + \mathbf{F}_m = \int_{A_m}\mathbf{q}dA + \mathbf{F}_m = \mathbf{0}, \tag{18}$$

$$\mathbf{T}_h + \mathbf{T}_m = \int_{A_m}\{\mathbf{q} \times (\mathbf{x} - \mathbf{x}_0)\}dA + \mathbf{T}_m = \mathbf{0} \tag{19}$$

where $\mathbf{x}_0$ is the hydrodynamic center of the particle, and $\mathbf{F}_m$ and $\mathbf{T}_m$ are external force and torque acting on $m$-th particle, respectively. The motion of the particles are described by 6 degrees of freedom: i.e., three translational velocities

$U = (U_x, U_y, U_z)$ and three rotational velocities $\boldsymbol{\Omega} = (\omega_x, \omega_y, \omega_z)$. Therefore, as the boundary condition, a given surface material point $\boldsymbol{x}_s$ on the particle moves with a velocity

$$\boldsymbol{v}(\boldsymbol{x}_s) = \boldsymbol{U} + \boldsymbol{\Omega} \times (\boldsymbol{x}_s - \boldsymbol{x}_0). \qquad (20)$$

The surface of each sphere is divided into $N_E = 512$ triangular elements and $N_N = 258$ nodes. According to Eqs. (17) and (20), the $i$-th node $\boldsymbol{x}_i$ on the particle surface has to satisfy a boundary condition

$$\boldsymbol{U} + \boldsymbol{\Omega} \times (\boldsymbol{x}_i - \boldsymbol{x}_0) + \frac{1}{8\pi\eta} \sum_m^M \left\{ \sum_e^{N_E} \boldsymbol{G}(\boldsymbol{x}_i, \boldsymbol{y}_e) \boldsymbol{q}(\boldsymbol{y}_e) \Delta A_e \right\} = \boldsymbol{0} \qquad (21)$$

where $\Delta A$ is the surface area of the element, subscript $e$ is the index of elements, and $\boldsymbol{y}_e$ is the position of the element $e$. The force- and torque-free conditions (18), (19) can be discretized as

$$\sum_e^{N_E} \boldsymbol{q}(\boldsymbol{x}_e) \Delta A_e = -\boldsymbol{F}_m, \qquad (22)$$

$$\sum_e^{N_E} \{\boldsymbol{q}(\boldsymbol{x}_e) \times (\boldsymbol{x}_e - \boldsymbol{x}_0)\} \Delta A_e = -\boldsymbol{T}_m. \qquad (23)$$

Note that four-point Gaussian quadrature is used to calculate the surface integral over each element. For singular elements, we work in polar coordinates to remove the $1/r$ singularity[28].

The external magnetic field is imposed in $xz$-plain, $\boldsymbol{B}^{ex}(t) = (B \cos(2\pi ft), 0, -B \sin(2\pi ft))$, and the torque acting on each particle can be obtained as

$$\boldsymbol{T}_m = \boldsymbol{m} \times \{\boldsymbol{B}^{ex} + \boldsymbol{B}_m^{dd}\} \qquad (24)$$

where $\boldsymbol{B}^{dd}$ is the magnetic field that is created by other particle:

$$\boldsymbol{B}_m^{dd} = \sum_{j \neq m} \frac{\mu_0}{4\pi r^3} \{3(\boldsymbol{m}_j \cdot \boldsymbol{n})\boldsymbol{n} - \boldsymbol{m}_j\} \qquad (25)$$

where $\mu_0$ is the vacuum permeability, $r$ is the particle distance, and $\boldsymbol{n}$ is the normal vector pointing from particle $m$ to $j$. We have three components for the external force: the gravitational force $\boldsymbol{F}_m^g$, the magnetic dipolar force $\boldsymbol{F}_m^{dd}$, and the repulsive force $\boldsymbol{F}_m^{rep}$, as follows

$$\boldsymbol{F}_m^g = -\frac{4}{3}\pi a^3 \Delta\rho g \widehat{\boldsymbol{z}}, \qquad (26)$$

$$\boldsymbol{F}_m^{dd} = \sum_{j \neq m} \frac{3\mu_0}{4\pi r^4} \{5(\boldsymbol{m}_j \cdot \boldsymbol{n})(\boldsymbol{m}_m \cdot \boldsymbol{n})\boldsymbol{n} - (\boldsymbol{m}_m \cdot \boldsymbol{n})\boldsymbol{m}_j - (\boldsymbol{m}_j \cdot \boldsymbol{n})\boldsymbol{m}_m - (\boldsymbol{m}_m \cdot \boldsymbol{m}_j)\boldsymbol{n}\}, \qquad (27)$$

$$\boldsymbol{F}_m^{rep} = \sum_{r < r_0} k(r - r_0)\boldsymbol{n} \qquad (28)$$

where $\widehat{\boldsymbol{z}}$ is a normal vector pointing $+z$ direction, $k$ is the spring constant, and $r_0$ is the natural length. The repulsive force is introduced in order to avoid overlap between sphere–sphere and sphere–wall, and the force is present only when the distance is less than the natural length.

The velocities are obtained by solving the linear equations $\boldsymbol{Ax} = \boldsymbol{b}$ with a known vector $\boldsymbol{b} = \{\boldsymbol{v}^\infty, -\boldsymbol{F}_m, -\boldsymbol{T}_m\}$ and an unknown vector $\boldsymbol{x} = \{\boldsymbol{q}, \boldsymbol{U}, \boldsymbol{\Omega}\}$, where $\boldsymbol{A}$ is the dense matrix of size $M(3N_N + 6)$ based on Eqs. (21)–(23). For details, see previous works[26,29,30].

## Data availability
The data that support the findings of this study are available from the corresponding author upon reasonable request (ptierno@ub.edu).

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

## Acknowledgements
We acknowledge Fernando Martinez-Pedrero for initial experiments. This project has received funding from the European Unions Horizon 2020 research and innovation program under Grant Agreement number 665440. H.M. and P.T. acknowledge support from the ERC Grant "ENFORCE" (Number 811234). P.T. acknowledges support from from MINECO (FIS2016-78507-C2, ERC2018-092827), DURSI (2017SGR1061), and Generalitat de Catalunya under Program "ICREA Acadèmia".

## Author contributions
H.M.-C. performed the experiments. F.M. and D.M. conducted the theoretical and computational part of the work. H.M.-C., F.M., D.M., R.G., and P.T. wrote the paper, discussed, and interpreted the results.

## Additional information

**Competing interests:** The authors declare nocompeting interests.

