## [Peer Review File · Nature Communications]

Reviewers' comments:

Reviewer #1 (Remarks to the Author):

In this work, the authors demonstrate that a population of colloidal particles can be assembled into large scale propelling carpets, and the vacancies in the carpet can be filled with particles detached from the back of it. The tread-milling dynamics is realised by increasing the field amplitude, and it depends on the balance between magnetic interactions and hydrodynamics among the particles. The displays of the experimental results is not hard to follow and the mathematical analysis is nicely performed. Overall, it is a piece of interesting work.

However, the same group has published a paper in PRA (2015): "Magnetic Propulsion of Self-Assembled Colloidal Carpets: Efficient Cargo Transport via a Conveyor-Belt Effect", in which the authors had demonstrated the formation, locomotion, and splitting/merging behaviour with passive and active manners of the colloidal carpets, as well as the targeted delivery of some cargos. The overlap between these two works cannot be ignored. Therefore, the authors shall clearly discriminate the current work from the the previous one. After the clarification of novelty, I will regard this work as a qualified one for Nature Communications. Some minor questions and comments for this work are listed, which may potentially improve the quality of this manuscript.

- The analysis on Eq.3 and 4, i.e. judging the pose of the particles by the equation solution, is well performed. But how is Eq.3 obtained? Please give more details or give references on this. What is the relationship between ζ and ζ_R ? Is it a typo? Meanwhile, what is the relationship between ω and ω_c ?
- How is the scalability of the formation, locomotion of a carpet and its tread-milling behaviour? In other word, when the number of the particles lower/higher than a certain threshold, will the behaviours become unaccessible?
- I am interested in the different behaviour when the physical parameter of the fluids changes, e.g. viscosity and salt ions. The authors may provide theoretical analysis or some experimental results?

Reviewer #2 (Remarks to the Author):

The manuscript "Tunable self-healing of magnetically propelling colloidal carpets" the authors describe the annealing of defects in a colloidal crystal of magnetic beads. The experimental results are backed up by theory. The results presented in the manuscript are very interesting, the language is good and the conclusions are well drawn and base on the experimental and theoretical findings. The manuscript is certainly worth publishing but the authors might take into account my considerations below. I really enjoyed reading the manuscript and it triggered a range of thoughts, which I think could be clarified and discussed in more depth.

In my opinion the authors could go further with respect to generalisation as well as clarity of the presented work:

Particles: If I saw it correctly the surface coating of the particles is not discussed. Are the particles coated? Are charges involved? How sticky are they and how might this change the hydrodynamic boundary condition or to be more precise is the viscosity of the water really the most important parameter?

Fields: The authors use a rotating in and out of plane field to move and anneal the magnetic beads. Even though it is somewhat mentioned in the manuscript I think the point is not made clear enough. For the in plane field the particles experience an attractive force in the plane. For the out of plane case the force is repulsive in the plane. This might even open up defects which can then be filled more easily.

"Phase diagram (Fig. 1 g)": This is a really nice result and one of the key panels of the paper as I see it. The authors discuss the regions with respect to eq. 4. Still I think the point can be made clearer, like indicated above. At large out of plane fields particles organised in sheets out of plane attract each other, whereas for large in-plane fields they remain in plane. In the intermediate region the particles experience a change in attractive force from in to out of plane and back in plane. Possibly, for very special cases it might even be possible to move lattice planes past each other by this approach.

Theory/simulation: Given the authors have a theory in place, I wonder whether they could calculate the phase formation by doing a computer simulation assuming hydrodynamics and rotating dipolar forces. This might allow to reproduce the data presented in Fig. 3 d and e. These data scatter quite significantly at the moment and it might be possible to find the scaling of the parameters with time from simulations. I understand that particle tracking might be challenging in the microscope images.

Fig. 3 c: I wonder whether the authors have an explanation for the increase in speed with increasing field? Is this a result of particles sticking to the substrate, gravity or multi-particle interaction? From a first thought I would have expected an increase of speed with field frequency but not necessarily with field strength.

In the conclusions the authors state: "we report exactly the opposite, the faster the propelling carpets are the more rapidly they form ordered structures". For a crystal I would expect better annealing for "higher temperatures" as long as the temperatures are far enough from a phase boundary. Can the results be seen in the context of Oswald ripening? On the other hand the authors apply fields of a specific frequency, which is very different from temperature adding white noise to a system. Have the authors thought along this line? Also the fields applied are more like steering a solution rather than adding random noise. Or might the annealing be determined by local structures, which do not break up statistically and are very stable once they are formed?

It might be good to give an AC susceptibility of the particles to show that the applied frequencies are far from resonances.

With kind regards,
Max Wolff

Reviewer #3 (Remarks to the Author):

The authors in their manuscript "Tunable self-healing of magnetically propelling colloidal carpets" explore population of colloidal rollers assembled by magnetic fields into large scale propelling carpets, can form perfect crystalline materials upon suitable balance between magnetism and hydrodynamics. Authors then demonstrate a field-tunable annealing protocol based on a particle flow above the carpet that enables complete crystallization after a few seconds of propulsion.

The authors previously reported the formation of the magnetic colloidal carpets (ref 20 in their manuscript) and the current focus is on the mechanism of self-healing of these active crystals. In their system it is facilitated by a colloidal flow of particles. Activity based annealing of colloidal crystals is not a novel phenomenon and has been demonstrated previously both in experiments and simulations

(for instance, Dijkstra group, Chaikin group, etc) in different systems. The annealing of active crystal is quite intuitive as one needs some energy to heal defects in the lattice, and this energy is naturally present in active particle systems. While I do not claim the reported system and observed phenomenon are not interesting, I do feel the present work lacks enough novelty to be published in Nature Communications and more suited for specialized journals.

Response to Reviewers' comments

We thank all the referees for their thorough reading of our manuscript and for the constructive suggestions/criticisms. Our point-by-point responses and the corresponding changes to the manuscript are described below.

Response to Reviewer #1

Reviewer: *In this work, the authors demonstrate that a population of colloidal particles can be assembled into large scale propelling carpets, and the vacancies in the carpet can be filled with particles detached from the back of it. The tread-milling dynamics is realised by increasing the field amplitude, and it depends on the balance between magnetic interactions and hydrodynamics among the particles. The displays of the experimental results is not hard to follow and the mathematical analysis is nicely performed. Overall, it is a piece of interesting work.*

Authors: We thank the referee for all the positive comments on our manuscript.

Reviewer: *However, the same group has published a paper in PRA (2015): "Magnetic Propulsion of Self-Assembled Colloidal Carpets: Efficient Cargo Transport via a Conveyor-Belt Effect", in which the authors had demonstrated the formation, locomotion, and splitting/merging behaviour with passive and active manners of the colloidal carpets, as well as the targeted delivery of some cargos. The overlap between these two works cannot be ignored. Therefore, the authors shall clearly discriminate the current work from the previous one.*

After the clarification of novelty, I will regard this work as a qualified one for Nature Communications.

Authors: We now clearly distinguish the current work from the previous one which was published in PRApplied 3, 051003 (2015). As the referee correctly state, the PRApplied demonstrates the formation of the carpet, its splitting/merging behaviour against one obstacle, and the transport of biological cargos. In the current manuscript we report the discovery of the self-healing property of the carpet, that arises when exploring a different range of field parameters respect to the work published in PRApplied. We then demonstrate that increasing the propulsion speed of the carpet (and thus the activity of the system), increases the ordering via a controlled colloidal flow, which is a counter-intuitive observation. Thus, we use the carpet as a model system to demonstrate a phenomenon that was not observed/reported previously. On the other hand, the PRApplied instead was centred on the application of the carpet as an efficient transporter of biological cargos. Note that these cargos randomly move above the carpet, and do not give rise any annealing of the system. To clarify this point in the manuscript, we write on page 1, column 2 the following:

"In a previous work [20], we demonstrate that the carpet could be used as an efficient drug-delivery vector for transporting biological cells across its surface. Beyond its technological applications, here we report the discovery that such carpet could be used as a general model system for nonequilibrium crystallization process, by demonstrating

a self-healing process that can be controlled by an external magnetic field. We show that by increasing the field amplitude the carpet displays, ”

[20] F. Martinez-Pedrero and P. Tierno, *Phys. Rev. Applied*, 3, 051003 (2015)

Further, to stress the novelty of the current work, we have added more experimental data and theoretical analysis focused on the observed phenomenon of self-healing. We measure the self-healing time (see new Fig.3(e)), the scalability of the system (see answer below and the new image in Supporting Information) and we analyse the dependence of the carpet speed on the field amplitude (see new Fig.3(c)). All these results strengthen our claims and make the article stand out with respect to the PRApplied. The only common point with the PRApplied is Fig.1(a) that illustrates the carpet formation and could appear similar for both articles. However, we decide to keep this image since we believe it should facilitate the reader's understanding of the carpet formation. This is the starting point to explain the new phenomenology reported in the manuscript.

Reviewer: *Some minor questions and comments for this work are listed, which may potentially improve the quality of this manuscript.*

- The analysis on Eq.3 and 4, i.e. judging the pose of the particles by the equation solution, is well performed. But how is Eq.3 obtained? Please give more details or give references on this.

Authors: We thank the Referee for this comment. In the previous version, we describe how Equation 3 was derived in detail in the Methods section. We reformulate the phrase to clarify this point in the main text, and write on page 3, column 1 the following:

“By combining the contributions from the magnetic dipole-dipole interaction and hydrodynamics, the effective total energy of the colloid that is located at the edge becomes $U_{\text{tot}} = U_{\text{m}} + U_{\text{h}}$. This leads to the...”

Reviewer: *What is the relationship between ζ and ζ_R ? Is it a typo?*

Authors: We thank the referee for noticing this typo, since both symbols ζ and ζ_R refer to the same friction coefficient. Thus, we use ζ all over the text.

Reviewer: *Meanwhile, what is the relationship between ω and ω_c ?*

Authors: We write in the main text the relationship between ω and ω_c and refer to the Method section where we give a detailed derivation of the expression of ω_c . We write on page 2, column 1 the following:

“Here $\omega_c = B_0^2 \chi \tau_r \omega / 6 \eta \mu_0 (1 + \tau_r^2 \omega^2)$ is the particle angular velocity, which is in general lower than the driving one, $\omega = 2 \pi f$. Here τ_r is the magnetic relaxation time of the paramagnetic colloids, see Methods.”

Reviewer: - How is the scalability of the formation, locomotion of a carpet and its tread-milling behavior? In other word, when the number of the particles lower/higher than a certain threshold, will the behaviors become unaccessible?

Authors: The question raised by the referee triggered us to perform further experimental analysis to investigate the scalability of the carpet formation, propulsion and tread-milling. In these new experiments we have found that:

1) for the scalability of the carpet formation, above a minimum carpet size formed by $N \sim 150$ particles, the carpet can be formed with complete order and no annealing is required. This is because this size is approximately equal to a grain boundary. To determine this critical size, we have measure the bond orientational order parameter of different carpets assembled by the rotating field. The results are shown in the image below, with the continuous line being an average of the experimental data (scattered points).

2) Regarding the scalability of the carpet propulsion with the size, we find similar results as the one reported in Phys. Rev. Applied, 3, 051003 (2015). The results are shown in the image below where we measure the average speed of different carpets and show it in semi-log plot. This speed initially rapidly increases with N , while it saturates around $N \sim 300$, in agreement with what was previously observed.

3) Finally, regarding the treadmilling behaviour, we observe that this effect occurs for all the carpet sizes, however for carpet composed by less than $N = 150$ particles, we did not observe any annealing, since for such small size the carpet usually assemble already with crystalline order. This can be also appreciated by the image at the bottom, where we measure the orientational bond order parameter versus time for different carpets, until the maximum size of $N= 4000$ particles that completely fills the observation area.

We place these three images in the supporting information (Figures S1, S2 and S3) and comment on these points in the manuscript on page 5, column 1 by writing the following:

“We also investigate the scalability of the carpet formation (Figure S2), propulsion (Figure S3) and the tread-milling motion (Figure S4). By varying the number of particles in the carpet, we find that the initial self-assembly process could lead to an ordered crystalline structure below $N = 150$ particles, as shown in Figure S2 in the SI. Larger carpets

assembled by the rotating field display grain boundaries and vacancies that require the tread-milling motion to crystallize. The carpet speed increases with the number of particles, reaching a saturation value above $N \sim 300$ particles, as shown in Figure S3 in SI. Also, we found that the tread-milling behaviour treadmilling behaviour can be observed for all the carpet size, and until filling completely the observation area of our experimental system for $N = 4000$ particles; see Figure S4 in SI.”

Reviewer: - I am interested in the different behavior when the physical parameter of the fluids changes, e.g. viscosity and salt ions. The authors may provide theoretical analysis or some experimental results?

Authors: The referee has asked interesting questions on the effect of salt and viscosity on the system dynamics. The addition of salt reduces the thickness of the double layer, favouring irreversible aggregation between the particles due to attractive van der Waals interactions. The paramagnetic colloids are doped with iron-oxide nanoparticles that make their Hamaker constant larger than simple non-magnetic particles. We have checked the effect of salt by repeating the experiments with the addition of 0.15M of NaCl and found that in some cases the particles stick to the surface showing no propulsion. We attach a video that illustrate this behaviour, see video ForReferee1.AVI.

Regarding the role of the viscosity η , since the system dynamics is overdamped, increasing or decreasing η slows down or speeds up the process. In other words, the viscosity of the system only sets a time scale. We demonstrate this point by performing further experiments where we have propelled a carpet in water and inside a mixture of water/glycerol (50/50%) that increases the medium viscosity to $\eta = 6 \cdot 10^{-3}$ Pa s. We measure a corresponding decrease in the carpet speed from 1.8 micron/s to 0.7 micron/s as shown in the following image:

Response to Reviewer #2

Reviewer: *The manuscript "Tunable self-healing of magnetically propelling colloidal carpets" the authors describe the annealing of defects in a colloidal crystal of magnetic beads. The experimental results are backed up by theory. The results presented in the manuscript are very interesting, the language is good and the conclusions are well drawn and base on the experimental and theoretical findings. The manuscript is certainly worth publishing but the authors might take into account my considerations below. I really enjoyed reading the manuscript and it triggered a range of thoughts, which I think could be clarified and discussed in more depth.*

Authors: We thank the Reviewer for all the positive comments on our manuscript. We have considered all the recommendations and improved the manuscript accordingly.

Reviewer: *In my opinion the authors could go further with respect to generalisation as well as clarity of the presented work:*

Particles: If I saw it correctly the surface coating of the particles is not discussed. Are the particles coated? Are charges involved? How sticky are they and how might this change the hydrodynamic boundary condition or to be more precise is the viscosity of the water really the most important parameter?

Authors: The paramagnetic colloids are coated with surface carboxylic groups, and in high deionized water (low salt density), such groups dissociate to generate a negative charge and thus a double layer. In water the paramagnetic colloids sediment to a glass bottom plate, and once there they float a few nanometres above it due to electrostatic interactions with its surface (here a charged glass plate). We have now added this note in the Methods section, about the surface coating of the paramagnetic colloids:

"The particles are coated with surface carboxylic acid group with an active chemical functionality of $150 \mu\text{mol} \cdot \text{g}^{-1}$ per particle. When dispersed in highly deionized water ($18.2 \text{M}\Omega \cdot \text{cm}$, MilliQ system), hydrogen ions (H^+) dissociate from such groups leaving a negative charged surface and inducing the formation of a double layer."

Instead of:

"The particles are dispersed in highly deionized water (18.2 M Ω , MilliQ system)."

The questions regarding the stability of the colloidal particles and the role of viscosity in the dynamics of the system are like those posed by Reviewer #1; please see above for the corresponding replies.

Reviewer: *Fields: The authors use a rotating in and out of plane field to move and anneal the magnetic beads. Even though it is somewhat mentioned in the manuscript I think the point is not made clear enough. For the in plane field the particles experience and attractive force in the plan. For the out of plane case the force is repulsive in the plane. This might even open up defects which can then be filled more easily.*

Authors: The referee is right to state that for static in-plane (out of plane) fields the particles experience attractive (repulsive) interactions. However, in the formation of the colloidal carpet and in its propulsion, we use time-dependent magnetic fields with relatively high rotational frequency ($f > 20\text{Hz}$). Under such conditions the paramagnetic colloids experience time-averaged dipolar interactions that works as follows: for a rotating field in the substrate $[(x,y)]$ plane the particles are on average attractive towards each other along the plane, but repulsive along the z direction. For a rotating field in the (x,z) plane, the particles are attractive along the plane and thus they form elongated structures, while they are repulsive along the y -direction. We clarify better this point in the manuscript and write on page 1, end of column 1 the following:

“For sufficiently high frequencies, the rotating field induces attractive dipolar interactions that are isotropic when time-averaged~\cite{Mar15}. The dipolar interaction between two equal dipoles $\mathbf{m}_{\{i,j\}}$ at a distance $\mathbf{r}_{\{ij\}} = \mathbf{r}_{\{i\}} - \mathbf{r}_{\{j\}}$ is given by $U_m = -\mu_0/4\pi \{ [3(\mathbf{m}_{\{i\}} \cdot \mathbf{r}_{\{ij\}})(\mathbf{m}_{\{j\}} \cdot \mathbf{r}_{\{ij\}})/r^5 - (\mathbf{m}_{\{i\}} \cdot \mathbf{r}_{\{ij\}})/r^3] \}$, and becomes maximally attractive (repulsive) for particles with magnetic moments parallel (normal) to $\mathbf{r}_{\{ij\}}$. Performing a time average of the potential gives an effective attractive interaction in this plane $\langle U_m \rangle = -[\mu_0 m^2/8\pi (x+y)]^3$.”

Reviewer: *“Phase diagram (Fig. 1 g)”: This is a really nice result and one of the key panel of the paper as I see it. The authors discuss the regions with respect to eq. 4. Still I think the point can be made clearer, like indicated above. At large out of plane fields particles organised in sheet out of plane attract each other, whereas for large in-plane field they remain in plane. In the intermediate region the particles experience a change in attractive force from in to out of plane and back in plane. Possibly, for very special cases it might even be possible to move lattice planes past each other by this approach.*

Authors: We thank the referee for these positive comments. We effectively observe cases where lattice planes can be move past each other with our annealing method. We attach a video that shows this possibility (ForReferee2.AVI) and comment on this point in the manuscript by writing on page 2, end of column 2:

“We also find that, for a certain range of field amplitudes close to the transition region in Fig. 1(g), our method allows the relative motion of two lattice planes of colloidal particles. This phenomenon could offer new possibilities in the study of frictional effects at the microscale in presence of hydrodynamic lubrication.”

Reviewer: *Theory/simulation: Given the authors have a theory in place, I wonder whether they could calculate the phase formation by doing a computer simulation assuming hydrodynamics and rotating dipolar forces. This might allow to reproduce the data presented in Fig. 3 d and e. These data scatter quite significantly at the moment and it might be possible to find the scaling of the parameters with time from simulations. I understand that article tracking might be challenging in the microscope images.*

Authors: We have initially planned large scale numerical simulations of the system using dipolar and hydrodynamic interactions, with the latter being considered in the far field approximation. However, we find that this approach did not capture the complexity of the

hydrodynamic flow generated by the ensemble of rotors near their surface. Thus, we decided to perform Boundary Element simulations, as described in the Methods section. With this approach we were able to resolve the particle shape and the full hydrodynamic flow field, as shown in Fig.2(c) of the manuscript. due to substantially higher computational cost, it was difficult to extend the simulations to the large carpet sizes as used in Fig. 3(d).

Reviewer: *Fig. 3 c: I wonder whether the authors have an explanation for the increase in speed with increasing field? Is this a result of particles sticking to the substrate, gravity or multi-particle interaction? From a first thought I would have expected an increase of speed with field frequency but not necessarily with field strength.*

Authors: The increase of the carpet speed with the amplitude of the magnetic field is due to the dependence of the particle rotational frequency ω_c on the square of the field amplitude, $\omega_c \sim B_0^2$. We indeed calculate the dependence of the carpet velocity with the field amplitude and use the obtained expression (see the new Methods section “Carpet speed”), to fit the experimental data shown in Fig.3(c). We find a good agreement between the theory and the experimental data, and a reasonable value of the obtained constant. We explain this point in the main text by writing on page 5, column 1 the following:

“The carpet speed can be calculated by considering a triangular lattice of surface rotors, with lattice constant a ; see Methods section for more details. We can fit the experimental data of Fig. 3(c) with Eq.~\ref{carpetspeed} using $\sim 3.2 \sqrt{\mu m}$ with the magnetic relaxation time τ_r as an adjustable parameter. We obtain a value of $\tau_r = 0.3 \cdot 10^{-4}$ s that is consistent with the diagram in Fig.1(g).”

See also the Method section for the detailed derivation of the carpet speed.

Reviewer: *In the conclusions the authors state: “we report exactly the opposite, the faster the propelling carpets are the more rapidly they form ordered structures”. For a crystal I would expect better annealing for “higher temperatures” as long as the temperatures are far enough from a phase boundary. Can the results be seen in the context of Oswald ripening? On the other hand the authors apply fields of a specific frequency, which is very different from temperature adding white noise to system. Have the authors thought along this line? Also the fields applied are more like steering a solution rather than adding ransom noise. Or might the annealing be determined by local structures, which do not break up statistically and are very stable once they are formed?*

Authors: The referee raises a number of very interesting points about possible analogies. We agree with the referee that the mechanism for annealing in our system is closer to shaking or stirring of the system at a chosen frequency than Oswald ripening by “thermal” fluctuations or white noise. However, the specific mechanism is subtler as there are feedback mechanisms coupling internal ordering to guiding of transported particles that will fill vacancies with the help of attractive dipolar interactions. We have decided that such analogies or their generalizations deserve careful and more thorough considerations and we have relegated such studies to a future publication. We would like to thank the referee for these good suggestions.

Reviewer: *It might be good to give an AC susceptibility of the particles to show that the applied frequencies are far from resonances.*

Authors: The AC magnetic susceptibility of these paramagnetic particles (Dynabeads M-270) was measured by X. J. A. Janssen et al. (Biosensors and Bioelectronics 24, 1937, 2009), where the authors found resonance around 400 kHz, thus well away from the experimental frequencies. We comment on this point in the text, and write:

“We use a narrow frequency range $\nu \in [20, 100] \text{ Hz}$, far away from resonance frequency ($\sim 400 \text{ kHz}$) as reported in the past [New Ref].”

[New Ref.] X. J. A. Janssen, A. J. Schellekens, K. van Ommering, L. J. van Ijzendoorn, and M. J. Prins, Controlled torque on superparamagnetic beads for functional biosensors. Biosens. Bioelectron. 24, 1937 (2009).

Response to Reviewer #3

Reviewer: *The authors in their manuscript "Tunable self-healing of magnetically propelling colloidal carpets" explore population of colloidal rollers assembled by magnetic fields into large scale propelling carpets, can form perfect crystalline materials upon suitable balance between magnetism and hydrodynamics. Authors then demonstrate a field-tunable annealing protocol based on a particle flow above the carpet that enables complete crystallization after a few seconds of propulsion.*

The authors previously reported the formation of the magnetic colloidal carpets (ref 20 in their manuscript) and the current focus is on the mechanism of self-healing of these active crystals. In their system it is facilitated by a colloidal flow of particles. Activity based annealing of colloidal crystals is not a novel phenomenon and has been demonstrated previously both in experiments and simulations (for instance, Dijkstra group, Chaikin group, etc) in different systems.

Authors: We agree with the referee that activity-based annealing of colloidal crystals has been reported recently via numerical simulation by the group of M. Dijkstra when mixing few active particles to a bath of passive ones. We apologize for missing the reference to this work, and now give credit to it in the main text, by writing on page 1, column 2:

“We also note that a crystallization process induced by a few active dopants in a bath of passive particles has been demonstrated recently in computer simulation studies [New Refs].”

[New Refs]

- R. Ni, A. C. S. Cohen, M. Dijkstra, Pushing the glass transition towards random close packing using self-propelled hard spheres. Nat Commun. 4, 2704 (2013).

- R. Ni, A. C. S. Cohen, M. Dijkstra, P. G. Bolhuis Crystallizing hard-sphere glasses by doping with active particles. Soft Matter 10, 6609-6613 (2013).

However, to our knowledge, there are no experimental results of annealing processes with active particle systems, since all the existing works dealing with dense suspensions

have reported the formation and disaggregation of clusters over time. For instance, the work of the group of P. Chaikin, probably the Reviewer is referring to *Science* 339, 936-40 (2013), demonstrates the formation of clusters of active particles that dynamically break and reform with large fluctuations, but not annealing of their internal order. Other works such as the one from the group of C. Bechinger (*Soft Matter* 21, 6187 (2015)), C. Bizzone (*Nat. Commun.* 2018; 9; 696 (2018)) or L. Isa (*Phys. Rev. Lett.* 120, 268004 (2018)) also investigate the statistics of cluster formation and disaggregation, as well the melting due to active dopants. However, they did not report annealing effects. We also cite the latter article in the manuscript. We add the following phrase on page 1, column 2:

“Recent experiments with active particles report melting [New Ref.1], clustering [New Refs. 2,3] or interstitial dynamics [New Ref. 4] but not annealing.”

*[New Refs 1] F. Klumpp, P. Shabestari, C. Lozano, G. Volpe, C. Bechinger, Formation, compression and surface melting of colloidal clusters by active particles *Soft Matter* 11, 6187 (2015).*

*[New Refs 2] J. Palacci, S. Sacanna, A. P. Steinberg, D. J. Pine, P. M. Chaikin, Living crystals of light-activated colloidal surfers. *Science*. 339, 936 (2013).*

*[New Refs 3] F. Ginot, I. Theurkauff, F. Detcheverry, C. Ybert, C. Cottin-Bizonne Aggregation-fragmentation and individual dynamics of active clusters *Nat. Comm.* 9, 696 (2018).*

*[New Refs 4] Kilian Dietrich, Giovanni Volpe, Muhammad Nasruddin Sulaiman, Damian Renggli, Ivo Buttinoni, and Lucio Isa Active Atoms and Interstitials in Two-Dimensional Colloidal Crystals *Phys. Rev. Lett.* 120, 268004 (2018).*

Reviewer: *The annealing of active crystal is quite intuitive as one needs some energy to heal defects in the lattice, and this energy is naturally present in active particle systems.*

Authors: We agree with the referee that energy is required to heal defects in crystalline materials. However, the mechanism of annealing in our system constitutes a fundamentally new process. In a typical annealing process, atoms migrate in the crystal lattice due to thermal heating and this migration decreases the number of dislocations, leading to a change in ductility and hardness. On the other hand, in driven and active systems, increasing the activity, i.e. the propulsion speed, usually leads to the melting of the system rather than annealing. Here, we use a driven system to demonstrate a novel mechanism for annealing, not based on a generalization of the annealing as induced by thermal fluctuations as noted by Reviewer #2, but on a fine balance between magnetism and hydrodynamics. With this mechanism we observe a novel scenario for annealing, in which increasing the speed of the carpet reduces the annealing time and makes the structure more ordered.

Reviewer: *While I do not claim the reported system and observed phenomenon are not interesting, I do feel the present work lacks enough novelty to be published in *Nature Communications* and more suited for specialized journals.*

Authors: To address the concern of the Referee, we have made a number of changes. In particular, we have improved the manuscript by:

- (i) Stating clearly the novelty of the current work with respect to our previous manuscript published in PRApplied. Please also see answer to Reviewer #1
- (ii) Providing more experimental data to demonstrate the scalability of the carpet, and the self-healing time, please see also the answer to Reviewer #1
- (iii) Providing further theoretical analysis of the system based on calculating the average speed of the carpet in the triangular geometry and the self-healing time; see the corresponding answer to Reviewer #2.

With these additions, and the complete revision of the text, we believe the manuscript improved in quality and can become suitable for Nature Communications.

REVIEWERS' COMMENTS:

Reviewer #1 (Remarks to the Author):

The authors have addressed all of my questions properly. I have no further questions and the manuscript can be published in the current form. Good work to the authors.

Reviewer #2 (Remarks to the Author):

Dear Editor,

the authors have addressed all questions raised by me earlier and I can recommend publication of the manuscript.